# The interface between SARS-CoV-2 and non-communicable diseases (NCDs) in a high HIV/TB burden district level hospital setting, Cape Town, South Africa

Ayanda Trevor Mnguni[1,2], Denzil Schietekat[2], Nabilah Ebrahim[2], Nawhaal Sonday[2], Nicholas Boliter[2], Neshaad Schrueder[1], Shiraaz Gabriels[1], Annibale Cois[3,4], Jacques L. Tamuzi[3], Yamanya Tembo[4], Mary-Ann Davies[4,5], Rene English[3], Peter S. Nyasulu[3,6]*

1 Department of Medicine, Faculty of Medicine and Health Sciences, Stellenbosch University, Cape Town, South Africa, 2 Khayelitsha District Hospital, Cape Town, South Africa, 3 Department of Global Health, Faculty of Medicine and Health Sciences, Stellenbosch University, Cape Town, South Africa, 4 School of Public Health and Family Medicine, University of Cape Town, Cape Town, South Africa, 5 Health Impact Assessment Directorate, Western Cape Government, Cape Town, South Africa, 6 Division of Epidemiology & Biostatistics, School of Public Health, Faculty of Medicine and Health Sciences, University of the Witwatersrand, Johannesburg, South Africa

* pnyasulu@sun.ac.za

## Abstract

### Background

COVID-19 experiences on noncommunicable diseases (NCDs) from district-level hospital settings during waves I and II are scarcely documented. The aim of this study is to investigate the NCDs associated with COVID-19 severity and mortality in a district-level hospital with a high HIV/TB burden.

### Methods

This was a retrospective observational study that compared COVID-19 waves I and II at Khayelitsha District Hospital in Cape Town, South Africa. COVID-19 adult patients with a confirmed SARS-CoV-2 polymerase chain reaction (PCR) or positive antigen test were included. In order to compare the inter wave period, clinical and laboratory parameters on hospital admission of noncommunicable diseases, the Student t-test or Mann-Whitney U for continuous data and the X2 test or Fishers' Exact test for categorical data were used. The role of the NCD subpopulation on COVID-19 mortality was determined using latent class analysis (LCA).

### Findings

Among 560 patients admitted with COVID-19, patients admitted during wave II were significantly older than those admitted during wave I. The most prevalent comorbidity patterns were hypertension (87%), diabetes mellitus (65%), HIV/AIDS (30%), obesity (19%), Chronic Kidney Disease (CKD) (13%), Congestive Cardiac Failure (CCF) (8.8%), Chronic

**Data Availability Statement:** All relevant data are within the paper and its Supporting information files.

**Funding:** The authors received no specific funding for this work.

**Competing interests:** The authors received no specific funding for this work.

**Abbreviations:** ACE2, Angiotensin-Converting Enzyme 2; AIDS, acquired immunodeficiency syndrome; AKI, Acute kidney injury; ARDS, Acute respiratory distress syndrome; ART, antiretroviral therapy; BMI, body mass index; CHD, chronic heart disease; CKD, chronic kidney diseases; COPD, Chronic obstructive pulmonary disease; COVID-19, CCF: congestive cardiac failure; Coronavirus disease 2019; Cr, creatine; CRP, C-Reactive protein; CVDs, cardiovascular diseases; CXR, chest x-ray; DIC, Disseminated intravascular coagulation; FIO2, fraction of inspired oxygen; HAP, Hospital-acquired; HGT, hemo glucose test; ICU, intensive care unit; IHD, ischemic heart disease; IL-6, Interleukin-6; IQR, interquartile ranges; JHU, John Hopkin University; LCA, Latent Class Analysis; NCDs, non-communicable diseases; pneumonia HIV, human immunodeficiency virus; RT-PCR, reverse transcriptase polymerase chain reaction; SARS-CoV-2, severe acute respiratory syndrome coronavirus 2; SE, standard deviation; SPO2, Oxygen saturation; TB, tuberculosis; TDF, Tenofovir disoproxil fumarate; TNF, tumor necrosis factor; WCC, white cell count.

Obstructive Pulmonary Disease (COPD) (3%), cerebrovascular accidents (CVA)/stroke (3%), with similar prevalence in both waves except HIV status [(23% vs 34% waves II and I, respectively), p = 0.022], obesity [(52% vs 2.5%, waves II and I, respectively), p <0.001], previous stroke [(1% vs 4.1%, waves II and I, respectively), p = 0.046]. In terms of clinical and laboratory findings, our study found that wave I patients had higher haemoglobin and HIV viral loads. Wave II, on the other hand, had statistically significant higher chest radiography abnormalities, fraction of inspired oxygen (FiO2), and uraemia. The adjusted odds ratio for death vs discharge between waves I and II was similar (0.94, 95%CI: 0.84–1.05). Wave I had a longer average survival time (8.0 vs 6.1 days) and a shorter average length of stay among patients discharged alive (9.2 vs 10.7 days). LCA revealed that the cardiovascular phenotype had the highest mortality, followed by diabetes and CKD phenotypes. Only Diabetes and hypertension phenotypes had the lowest mortality.

## Conclusion

Even though clinical and laboratory characteristics differed significantly between the two waves, mortality remained constant. According to LCA, the cardiovascular, diabetes, and CKD phenotypes had the highest death probability.

## Introduction

The Coronavirus disease 2019 (COVID-19) pandemic continues to accelerate with South Africa at the time of writing experiencing its 4th wave of COVID-19 infections. As of 29th October 2022, official statistics report 4,027,157 cumulative confirmed cases of COVID-19, including 102,311 reported deaths in South Africa [1]. South Africa is a middle-income country with coinciding epidemics of non-communicable diseases (NCDs) and chronic infectious diseases including a high prevalence of human immunodeficiency virus (HIV) and tuberculosis (TB). Demographic and Health Survey found that 41% of adult women and 11% of men were obese, 46% of women and 44% of men were hypertensive, and 13% of women and 8% of men had diabetes [2] and estimated overall HIV prevalence is 13.7% [3] with 852 cases (95% CI 679–1026) per 100 000 population of TB [4]. Evidence suggests that COVID-19 patients with NCDs and chronic infectious diseases such as diabetes, hypertension, cardiovascular disease, chronic kidney diseases (CKD), HIV, and TB are at an increased risk of disease severity and mortality [5–7]. The most common comorbidities reported in high HIV/TB burden settings among patients with severe COVID-19 are hypertension, heart failure, diabetes, cancer in the previous 5 years, chronic pulmonary disease, obesity, CKD, HIV and TB [7–10]. While high-income countries have vaccinated a higher proportion of their populations, low-middle income countries lag due to a combination of factors such as vaccine inequity, slow vaccine rollout programs due to poor administration, and vaccine hesitancy. These comorbidities are still an urgent threat to COVID-19 severity and mortality in high HIV/TB and under-resourced health settings such as district hospitals. The Beta variant was identified as the primary cause of the rapid increase in infections during Wave II in South Africa [11, 12]. The increased disease severity, the ability to escape previously acquired immunity as evidenced by increased hospitalizations and case fatality rate were attributed to the various mutations in the Beta variant when compared to its predecessors such as the Alpha variant and the original wild-type Wuhan strain [12–14], and the ability to escape previously acquired immunity [12, 15]. This contributed to the severity and mortality of COVID-19 in wave II. District level

hospital experiences provide a unique opportunity to study the effects of the pandemic at the 'grassroots' level, which informs the success or failure of public health interventions. According to a review of the literature, there is no study describing the effects of NCDs on COVID-19 outcomes at a high burden HIV/TB district level hospital setting. The aim of this study is to investigate the comorbidities that were linked to COVID-19 severity and mortality between waves I and II.

## Methods

### Study design

This was a retrospective observational study on the epidemiological and clinical characteristics of COVID-19 at Khayelitsha District Hospital, Cape Town, South Africa from March 2020 – January 2021.

### Study population

We included all consecutive patients, 18 years and older with COVID-19 as confirmed by a positive antigen test or severe acute respiratory syndrome coronavirus 2 (SARS-CoV-2) reverse transcriptase polymerase chain reaction (RT-PCR) result requiring hospital admission from March 2020 until January 2021. The main indication for hospitalization was COVID-19 pneumonia requiring oxygen therapy. Patients were followed up until completed hospital course (either discharge, transfer to tertiary or field hospital or death). Patients with incomplete outcome data were excluded.

### Setting

South Africa has a dual health system that includes a publicly funded district health system that serves approximately 84% of the population nationally and a private health system that is primarily funded by private health insurance schemes [16]. The district level healthcare system, along with the primary healthcare system, are the primary points of contact for COVID-19 patients and oversee providing healthcare to the vast majority of South Africans. Khayelitsha District Hospital is a 330-bed hospital in Mandela Park, Khayelitsha that opened in 2012. Khayelitsha is a township in South Africa, south-east of Cape Town. Most of the people (98.6%) are Black Africans [17]. It was constructed during the Apartheid era to enforce the Group Areas Act. Khayelitsha was intended to be isolated, situated on dune land with a high risk of seasonal flooding, and entirely residential, with no designated commercial or industrial zones [18]. Most of the population (55.6%) live in informal housing. Khayelitsha has South Africa's highest concentration of poverty and unemployment rate. Furthermore, Khayelitsha has the worst health indicators in Cape Town, with the highest rates of mortality for stroke, hypertension, and diabetes mellitus [19, 20].

### Data collection

We collected baseline, the demographic, clinical, laboratory and outcome data on hospital admission, from digital registries, with additional data captured from electronic medical records.

The baseline and clinical data collected included demographic information such as age and sex, symptoms on admission to hospital, presence of comorbidities including, hypertension, diabetes, overweight or obesity [defined as a body mass index (BMI) $\geq 25$ or $\geq 30$ kg/m$^2$ respectively or as documented by treating clinicians as the BMI was not captured for all patients, cardiac disease, chronic kidney disease, and active or previous history of TB.

Baseline arterial blood gas and laboratory values including severity indices were also captured. These included the partial pressure of arterial oxygen to fraction of inspired oxygen (P/F Ratio), the white cell count (WCC), the neutrophil to lymphocyte ratio (N/L ratio), serum creatine (Cr), glycated haemoglobin (HbA1c) and the C-Reactive protein (CRP), hemo glucose test (HGT), uremia, CD4 count, and viral load were captured if they were done up to a year prior to admission. The main outcome of the study was in-hospital death or survival to discharge.

## Statistical methods

Data were imported into R Statistical Software v. 4.0.5 (R Core Team, Vienna, Austria) for pre-processing and analysis. Sample characteristics were described as mean and standard deviation (SD) or median and interquartile ranges (IQR) for continuous variables, and frequency for categorical measures. Student t test and or Mann–Whitney U test were used to investigate differences in the distribution of continuous variables between sub-groups of patients. $X^2$ test or Fishers' Exact test were used to investigate differences in the distribution of categorical variables. Latent Class Analysis (LCA) was used to identify latent subpopulations with similar NCD patterns [21]. The Aalen-Johansen method was used to assess the competing risks of in-hospital mortality and discharge [22].

Uncertainty in the estimates was quantified by reporting their 95% confidence intervals (CI) or standard error (SE). To define statistical significance, a p-value of 0.05 was used as a cut-off. To deal with missing data, a pairwise deletion method was used.

## Ethical approval

Ethical approval for the study was obtained from the Stellenbosch University health research ethics committee (Ethics reference number: N20/05/020_COVID-19). Since the data were obtained from hospital records, a waiver of consent was requested and granted. The data were entered and analyzed anonymously, and no attempt was made to link the data to an individual identifier.

## Results

### Sample characteristics

A total of 580 patients with SARS-CoV-2 infection were admitted to the hospital during the study period. From those, we excluded the 19 cases with missing triage and/or outcome (death/discharge) date, and a case with missing age at admission. Of the remaining 560, 367 patients were admitted during the first wave of the pandemic (April-July 2020 in South Africa), and 193 during the second (November 2020 –January 2021). The distribution of admissions by triage date is shown in Fig 1. Demographic, lifestyle, and clinical characteristics of the sample are summarised in Table 1, separately by wave.

Patients admitted during wave II were significantly older than those admitted during wave I (2.6% vs 7.9% of subjects under 40, 45% vs 38% of subjects aged 60–69 years), and were 3 times more likely to be current drinkers. The comorbidity patterns were overall similar, with the notable exception of higher prevalence of HIV positive status (34% vs 23%) and previous stroke (4.1% vs 1.0%) in wave I. It should be highlighted that the high proportion of missing data on HIV status calls for caution in interpreting the first result. In contrast, a higher prevalence of obesity was observed in wave II (52% vs 2.5%).

The most frequent presenting complaint was shortness of breath (71% of patients across the two waves), cough (69%), fever (37%), myalgia (21%), sore throat (14%) and chest pain (11%).

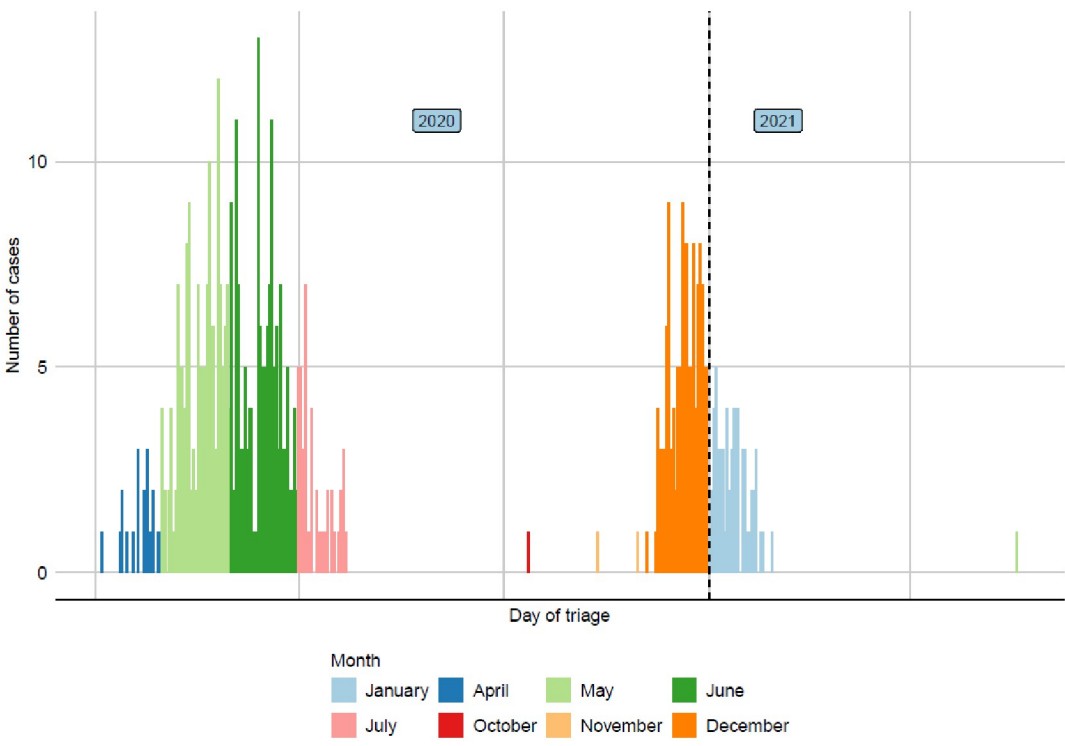

**Fig 1. Distribution of hospital admission by triage date.**

Shortness of breath and myalgia were significantly more frequent in wave II, and chest pain more common in wave I.

Table 2 summarises the data collected at triage and the results of the laboratory tests. Subjects in wave II had more frequently CXR abnormalities (55% vs 10%, p<0.001), higher median (IQR) temperature [36.80(36.10–37.10) vs 36.60(36.30–37.20), p = 0.014)] and $FiO_2$ [21(21–21) vs 21(21–80), p<0.001)] and lower systolic and diastolic BP [124 (117, 151) vs 133 (108, 137), p<0.001] and [76(68–83) vs 80(71–90), p<0.001] and pulse rate [97(88, 107) vs 102 (88–114), p = 0.004]. Laboratory results differed with higher mean uraemia in wave II [8(5, 13) vs 7(4–12), p = 0.033]. In contrast the median (IQR) haemoglobin and HIV viral load were higher in wave I [13.30(12.00–14.30) vs 12.70(11.50–13.90), p = 0.003] and [(20(20–28) vs 20 (20–20), p = 0.045)], respectively.

## Interventions and complications

Table 3 shows the frequency of various interventions in the two waves, while Table 4 summarises the reported complications.

## Interventions and complications

Table 3 shows the frequency of various interventions in the two waves, while Table 4 summarises the reported complications.

Most subjects had $O_2$, and corticosteroids administered during their hospital stay, and in both cases, the frequency of the intervention was significantly higher in wave II. The between-waves comparison of the type of antibiotics administered was partly hindered by the substantial number of cases where the type is unspecified. Overall, data suggests significantly higher

**Table 1. Demographic, lifestyle, and clinical characteristics of the sample.**

| Variable | Wave | | | p-value [(1)] |
|---|---|---|---|---|
| | Overall, N = 560 | Wave I, N = 367 | Wave II, N = 193 | |
| Gender | | | | 0.2 |
| Female | 374 (67%) | 238 (65%) | 136 (70%) | |
| Male | 186 (33%) | 129 (35%) | 57 (30%) | |
| Age | 58 (50, 66) | 57 (50, 66) | 59 (51, 68) | 0.092 |
| Age class (years) | | | | 0.032 |
| 18–39 | 34 (6.1%) | 29 (7.9%) | 5 (2.6%) | |
| 40–59 | 273 (49%) | 179 (49%) | 94 (49%) | |
| 60–79 | 228 (41%) | 140 (38%) | 88 (46%) | |
| >80 | 25 (4.5%) | 19 (5.2%) | 6 (3.1%) | |
| Current smoking | 17 (3.0%) | 9 (2.5%) | 8 (4.1%) | 0.3 |
| Current drug use | 1 (0.2%) | 0 (0%) | 1 (0.5%) | 0.3 |
| Current alcohol use | 14 (2.5%) | 5 (1.4%) | 9 (4.7%) | 0.023 |
| HIV/AIDS* | 130 (30%) | 96 (34%) | 34 (23%) | 0.022 |
| Current/Previous TB | 24 (4.3%) | 15 (4.1%) | 9 (4.7%) | 0.7 |
| Hypertension | 485 (87%) | 318 (87%) | 167 (87%) | >0.9 |
| Diabetes | 362 (65%) | 232 (63%) | 130 (67%) | 0.3 |
| Obesity | 109 (19%) | 9 (2.5%) | 100 (52%) | <0.001 |
| Dyslipidemia | 12 (2.1%) | 6 (1.6%) | 6 (3.1%) | 0.4 |
| CKD | 75 (13%) | 47 (13%) | 28 (15%) | 0.6 |
| Asthma | 10 (1.8%) | 8 (2.2%) | 2 (1.0%) | 0.5 |
| COPD | 17 (3.0%) | 9 (2.5%) | 8 (4.1%) | 0.3 |
| CCF | 49 (8.8%) | 26 (7.1%) | 23 (12%) | 0.054 |
| IHD | 10 (1.8%) | 9 (2.5%) | 1 (0.5%) | 0.2 |
| Mitral Valve Disease | 6 (1.1%) | 4 (1.1%) | 2 (1.0%) | >0.9 |
| Stroke | 17 (3.0%) | 15 (4.1%) | 2 (1.0%) | 0.046 |
| Cancers | 6 (1.1%) | 4 (1.1%) | 2 (1.0%) | >0.9 |
| Psychiatric disorders | 15 (2.7%) | 11 (3.0%) | 4 (2.1%) | 0.5 |
| Epilepsy | 8 (1.4%) | 7 (1.9%) | 1 (0.5%) | 0.3 |
| Other comorbidities | 66 (12%) | 28 (7.6%) | 38 (20%) | <0.001 |

n (%)

[(1)] Pearson's Chi-squared test; Fisher's exact test

Abbreviations: CKD: chronic kidney disease, CCF: congestive cardiac failure, COPD: Chronic Obstructive Pulmonary Disease, HIV/AIDS: human immunodeficiency virus/ acquired immune deficiency syndrome, IHD: Ischaemic Heart Disease, TB: tuberculosis

*: HIV status was only available for 428 of the 560 patients.

use of oxygen (72% vs 84%, P<0.001), corticosteroids (51% vs 94%, p<0.001), ceftriaxone (11% vs 27%, p<0.001), Azithromycin (9.5% vs 20%, p<0.001) in the second wave compared to the first. Acute Kidney Injury (19% vs 30%, p = 0.004) and ARDS (1.4% vs 21%, p<0.001) were more common complications in wave II, while shock (2.7% vs 0%, p = 0.018) was predominant in wave I.

## Outcomes

The proportion of subjects dead as a function of the number of days after triage is shown in Fig 2. The figure indicates a slightly higher risk of mortality in wave II, consistent over time.

**Table 2. Triage data and laboratory results.**

| Variable | N | Wave | | | p-value [1] |
|---|---|---|---|---|---|
| | | Overall, N = 560 | Wave I, N = 367 | Wave II, N = 193 | |
| CXR abnormality | 560 | 145 (26%) | 38 (10%) | 107 (55%) | <0.001 |
| Respiratory Rate | 545 | 20 (18, 28) | 20 (18, 28) | 20 (19, 26) | 0.4 |
| Pulse rate | 549 | 100 (88, 112) | 102 (88, 114) | 97 (88, 107) | 0.004 |
| Systolic BP | 547 | 129 (113, 148) | 133 (117, 151) | 124 (108, 137) | <0.001 |
| Diastolic BP | 547 | 79 (70, 87) | 80 (71, 90) | 76 (68, 83) | <0.001 |
| Hypertension | 547 | 208 (38%) | 160 (44%) | 48 (26%) | <0.001 |
| Temperature | 529 | 36.70 (36.20, 37.10) | 36.60 (36.10, 37.10) | 36.80 (36.30, 37.20) | 0.014 |
| SPO2 | 549 | 92 (85, 96) | 92 (85, 96) | 92 (84, 96) | 0.7 |
| PaO2 | 310 | 7.70 (6.10, 9.50) | 7.50 (6.10, 9.20) | 7.70 (6.30, 10.30) | 0.5 |
| FiO2 | 210 | 21 (21, 40) | 21 (21, 21) | 21 (21, 80) | <0.001 |
| HGT | 492 | 12 (7, 20) | 12 (7, 21) | 12 (8, 17) | 0.4 |
| Uremia | 553 | 7 (5, 13) | 7 (4, 12) | 8 (5, 13) | 0.033 |
| Creatinine | 553 | 93 (69, 145) | 94 (69, 143) | 93 (73, 158) | 0.5 |
| C-reactive protein | 491 | 148 (76, 229) | 146 (71, 231) | 154 (94, 224) | 0.3 |
| WCC | 552 | 9.2 (6.9, 11.8) | 9.2 (6.8, 12.1) | 9.4 (7.1, 11.5) | 0.8 |
| Hemoglobin | 551 | 13.10 (11.80, 14.15) | 13.30 (12.00, 14.30) | 12.70 (11.50, 13.90) | 0.003 |
| PLTS | 551 | 281 (220, 367) | 280 (215, 364) | 282 (225, 368) | 0.7 |
| Lymphocytes | 435 | 1.64 (1.25, 2.14) | 1.63 (1.19, 2.08) | 1.65 (1.29, 2.16) | 0.2 |
| Neutrophils | 434 | 6.7 (4.7, 8.9) | 6.6 (4.5, 9.0) | 6.8 (5.0, 8.6) | 0.8 |
| CD4 | 101 | 336 (212, 480) | 306 (228, 460) | 410 (147, 558) | 0.4 |
| Viral Load | 90 | 20 (20, 23) | 20 (20, 28) | 20 (20, 20) | 0.045 |

Median (IQR)

[1] Wilcoxon rank sum test

Abbreviations: BP; blood pressure, CD4: cluster of differentiation 4, HGT: Hemo Glucose Test, PLTS: Platelets, SpO2: Oxygen saturation, PaO2: Partial Pressure of Oxygen, FiO2: fraction of inspired oxygen, WCC: White Cell Count

However, the differences were not statistically significant, and disappeared completely after adjustment for age and comorbidities at baseline. The odds ratio for death vs discharge between wave I and wave II were 1.3 (95% CI: 0.76–2.19) in the unadjusted analyses, and 0.94 (0.84–1.05) after adjustment.

**Table 3. Interventions.**

| Variable | N | Wave | | | p-value [1] |
|---|---|---|---|---|---|
| | | Overall, N = 560 | Wave I, N = 367 | Wave II, N = 193 | |
| Co-trimoxazole | 560 | 6 (1.1%) | 3 (0.8%) | 3 (1.6%) | 0.4 |
| $O_2$ | 560 | 425 (76%) | 263 (72%) | 162 (84%) | 0.001 |
| Corticosteroids | 281 | 220 (78%) | 53 (51%) | 167 (94%) | <0.001 |
| Ceftriaxone | 560 | 93 (17%) | 40 (11%) | 53 (27%) | <0.001 |
| Azithromycin | 560 | 73 (13%) | 35 (9.5%) | 38 (20%) | <0.001 |
| Amoxicillin + Clavulanic Acid | 560 | 13 (2.3%) | 6 (1.6%) | 7 (3.6%) | 0.15 |
| Unspecified antibiotic | 560 | 323 (58%) | 278 (76%) | 45 (23%) | <0.001 |

n (%)

[1] Pearson's Chi-squared test; Fisher's exact test

**Table 4. Reported complications.**

| Variable | N | Wave | | | p-value [1] |
|---|---|---|---|---|---|
| | | Overall, N = 560 | Wave I, N = 367 | Wave II, N = 193 | |
| AKI | 560 | 126 (22%) | 69 (19%) | 57 (30%) | 0.004 |
| Shock | 560 | 10 (1.8%) | 10 (2.7%) | 0 (0%) | 0.018 |
| ARDS | 560 | 45 (8.0%) | 5 (1.4%) | 40 (21%) | <0.001 |
| HAP | 560 | 6 (1.1%) | 4 (1.1%) | 2 (1.0%) | >0.9 |
| DIC | 560 | 2 (0.4%) | 2 (0.5%) | 0 (0%) | 0.5 |

n (%)

[1] Pearson's Chi-squared test; Fisher's exact test

Abbreviations: AKI: Acute kidney injury, ARDS: Acute respiratory distress syndrome, DIC: Disseminated intravascular coagulation, HAP: Hospital-acquired pneumonia

The average survival time was 8 days in wave I (ranging from 0 to 58 days) and 6.1 days in wave II (1 to 25 days). The average length of stay of patients discharged alive was 9.2 days in wave I (ranging from 0 to 89 days) and 10.7 days in wave II (0 to 68 days).

## NCD patterns

Latent class analysis suggests the presence in the population of 4 latent classes. The comorbidity patterns observed in each class are shown in Fig 3. Phenotype 1 ('*Cardiovascular*') is

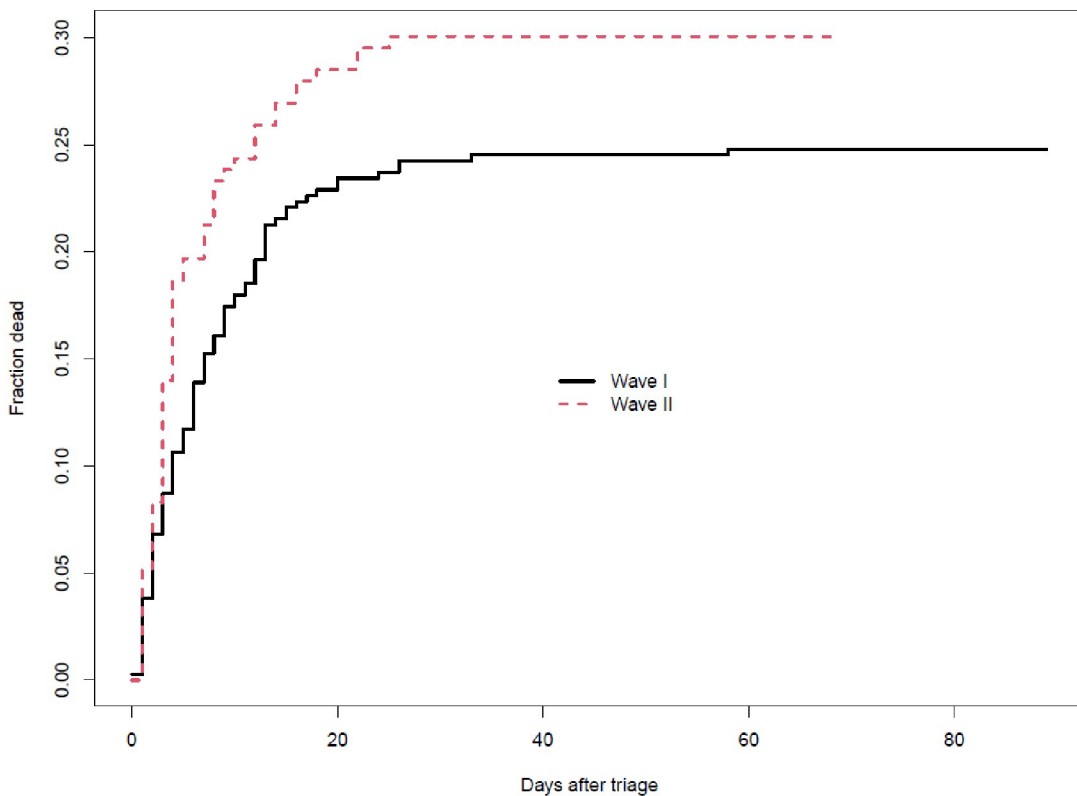

**Fig 2. Fraction dead vs. time from triage, by wave.**

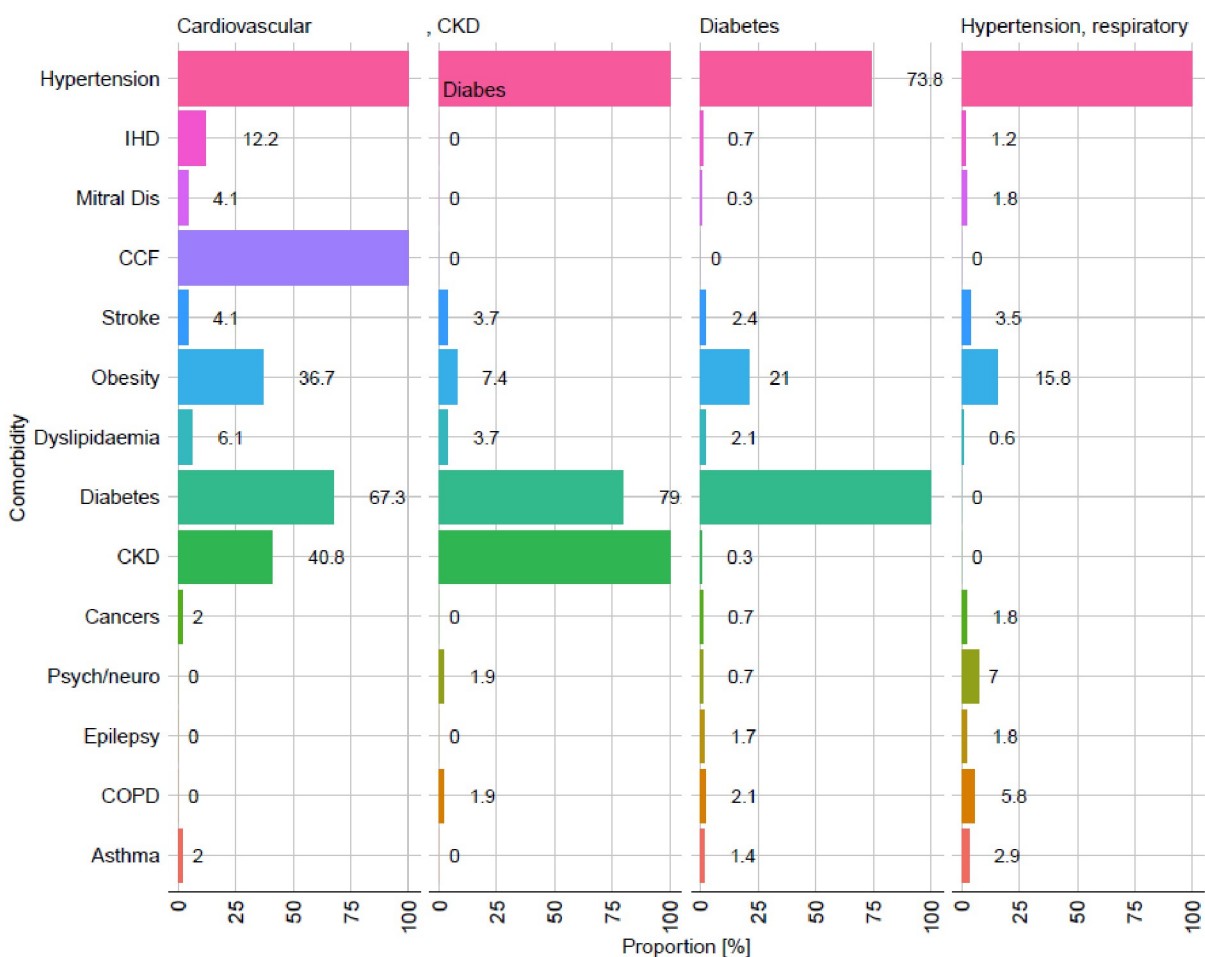

**Fig 3. Comorbid patterns by latent disease class.**

characterised by high prevalence of cardiovascular diagnoses (IHD, mitral valve disease, CCF, stroke, accompanied by significant prevalence of diabetes and CKD). Phenotype 2 ('*CKD*') is characterised by the highest prevalence of diabetes and CKD, but minimal prevalence of other diseases (excluding hypertension, which is highly prevalent across the four phenotypes). Phenotype 3 ('*Diabetes*') shows high prevalence of diabetes but no CKD and the lowest prevalence of obesity. Phenotype 4 ('Hypertension'), finally, shares with the remaining groups a high prevalence of hypertension but has a relatively 'healthy' comorbidity profile regarding the other disease considered.

Fig 4 shows the probability of deaths by time from triage, separately by latent phenotypes. It clearly indicated that individuals with phenotype 1 and 2 (Cardiovascular and CKD) have the highest risk of death at any point in time. Phenotype 3 and 4 (diabetes and hypertension) have the best probability of survival. The risk of death is similar for the two groups in the first two weeks from triage. Afterwards, phenotype "diabetes" had a slightly better prognosis.

## Discussion

This retrospective cohort study of 560 patients focused on the interactions between SARS-CoV-2 infection and NCDs during two COVID-19 waves in a district level hospital in South

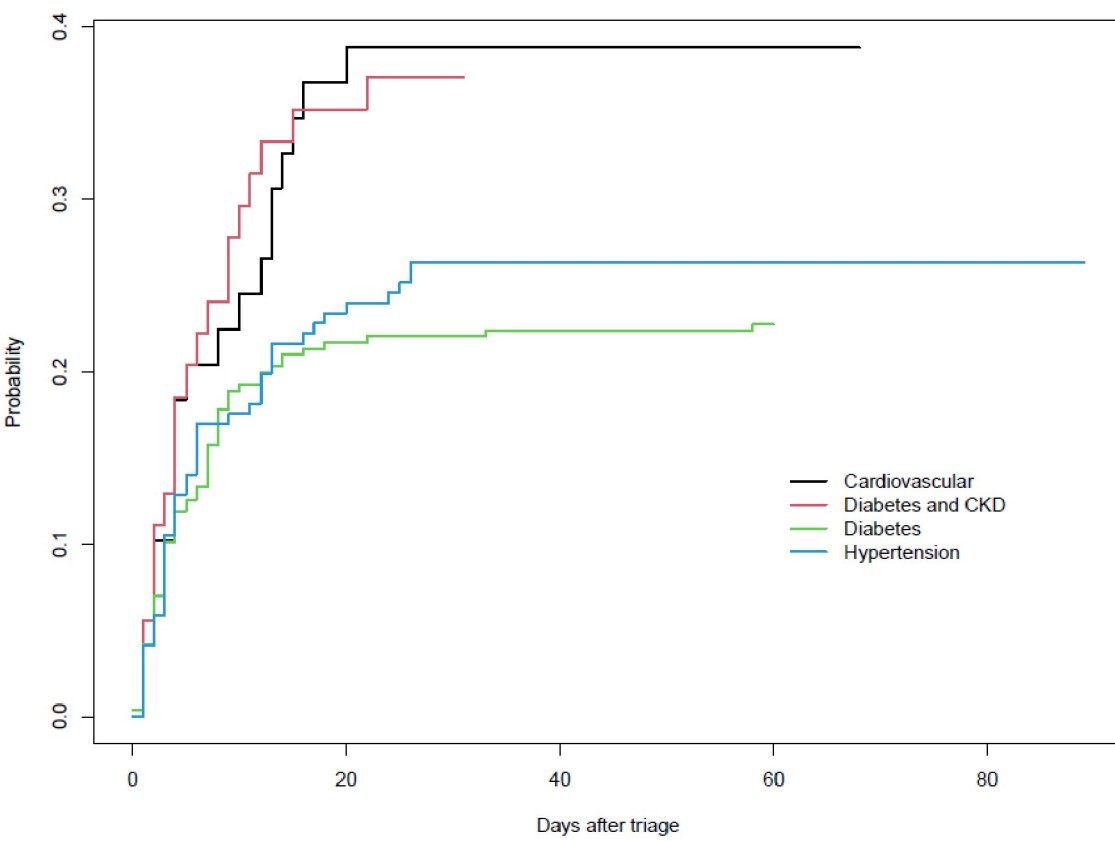

**Fig 4. Fraction dead vs time from triage, by latent class.**

Africa. The median (IQR) age was 58 (50–66) years and females (67%) were more prevalent than males (33%). The cohort included patients with hypertension (87%), diabetes (65%) and obesity (30%). A third of the cohort was HIV infected. The target organ damage associated with NCDs included CKD (13%), CCF (8.8%) and cerebrovascular accidents (3%). In comparison to wave II, wave I had a statistically significant higher proportion of patients with HIV/AIDS and stroke. In contrast, wave II had a statistically significant higher proportion of old age, current alcohol use, dyslipidemia, and CCF and obesity. The higher proportion of obesity in wave II should be interpreted with caution because there was little emphasis on capturing BMI in the first wave as obesity was not recognized as a marker of disease severity. Obesity/increased BMI was found to be associated with poor outcomes as the pandemic evolved. In terms of clinical and laboratory findings, our study found that wave I had a statistically significant higher pulse rate, systolic BP, diastolic BP, hypertension, haemoglobin, and HIV viral load. However, wave II had statistically significant higher CXR abnormalities, temperature, $FiO_2$, and uremia. In the latent class, our study found that cardiovascular disease, diabetes mellitus, and CKD had a higher mortality rate after triage than diabetes without CKD and hypertension. In wave I, the average survival time was 8.0 days, and in wave II, it was 6.1 days. Our findings were consistent with previous research that found shorter survival times in wave II [8, 11]. Lalla et al. demonstrated that the probability of intensive care unit (ICU) survival was higher during the second wave than the first [12]. This could be explained by evidence-based and effective available interventions such as High-flow nasal oxygen (HFNO) and steroids in the ICU setting. The average length of stay for discharged alive patients was 9.2 days

in wave I and 10.7 days in wave II. The data showed a slightly higher risk of mortality in wave II, but the difference was not statistically significant.

HIV was independently associated with increased COVID-19 mortality in a large cohort study conducted in the Western Cape [6]. Furthermore, people with HIV who had a history of unsuppressed viral load were more likely to die in a hospital than suppressed HIV patients [11]. Tenofovir disoproxil fumarate (TDF), which is used in the first-line antiretroviral therapy (ART) regimen in South Africa, has been linked to new or worsening renal failure [11, 23], and other antiretroviral drugs have been linked to hyperlipidaemia, cardiac disease, and diabetes [11, 24]. The presence of comorbidities among PLWH was linked to an increased risk of COVID-19 mortality [11]. Thus, HIV may play a significant role in the emergence of NCDs in high-burden settings like the one studied here. Additionally, higher HIV rates in wave I may be associated with non-communicable diseases, increasing the mortality rate in wave I as the adjusted odds ratio for death vs discharge between wave I and wave was 0.94. (0.84–1.05). In comparison to a large cohort that stated that current/previous TB were predictors of COVID-19 in-hospital mortality, the small number of current/previous TB cases (24/560, 4.3%) had no effect on the study findings [6, 7].

In our study, latent class analysis revealed that stroke was included in the cardiovascular (4.1%), CKD (3.7%), hypertension (3.5%), and diabetes mellitus (2.1%) phenotypes. Psychiatric disorders were also prevalent in the hypertension phenotype (1.8%). The presence of comorbidities was associated increased with severity of COVID-19 infection Cerebrovascular disease had the strongest association, followed by cardiovascular diseases (CVDs), chronic lung disease, diabetes, and hypertension [25]. Even though hypertension is associated with an up to 2.5-fold increased risk of severe or fatal COVID 19, particularly in older individuals compared to other comorbidities such as COPD (over 5-fold higher risk) [26, 27] and CKD (over 3-fold higher risk), it still has important clinical implications [27]. Hypertension is also associated with a pro-inflammatory state, as evidenced by increased levels of Angiotensin II, chemokines, cytokines, Interleukin-6 (IL-6) and tumour necrosis factor (TNF) [28]. According to one study, the risks of adverse heart failure outcomes increased significantly in patients with high systolic blood pressure, but this trend was less obvious in patients with high diastolic blood pressure [29]. Our findings were also consistent with a study that found high blood pressure to be a significant predictor of an unfavourable COVID-19 prognosis [29]. High SBP/DBP variability, on the other hand, was associated with a high risk of mortality, implying that maintaining stable in-hospital BP in COVID-19 patients is critical [29].

As previously discussed, evidence suggests that patients with noncommunicable diseases such as diabetes, hypertension, cardiovascular disease, and chronic kidney disease have increased Angiotensin-Converting Enzyme 2 (ACE2) receptor expression, which facilitates SARS-CoV-2 entry in the host and thus increases disease severity and mortality [5]. Obesity, metabolic complications, and COVID-19 severity were all linked, with a focus on fat mass distribution and insulin resistance [30]. This is well established in our latent analysis, where CCF (100%), diabetes mellitus (67.3%), CKD (40.8%), IHD (12.2%), and dyslipidaemia (6.1%) contributed to the highest cardiovascular phenotype mortality rate. Similarly, the phenotypes diabetes mellitus and CKD had a high mortality rate. Diabetic patients have an increased risk of COVID-19 due to increased expression of a type one protease called Furin [31]. Furin facilitates the attachment of the SARS-CoV-2 spike protein to the ACE-2 receptor in diabetic patients, which is associated with a dysregulated host immune response with increased ACE2 receptors and Furin levels, which may be associated with lower insulin levels and worsening inflammation [31]. Similarly, low-grade systemic inflammation associated with obesity is associated with poor outcomes in COVID-19 patients [32]. This low-grade inflammation is linked to a hyperimmune inflammatory state known as the cytokine storm [33]. The cytokine storm

that occurs during SARS-CoV-2 infection may also contribute to the high prevalence of anaemia found in our study. Haemoglobin levels decreased with age, as did the percentage of subjects with diabetes, hypertension, and other comorbidities [34].

In this study, we also found that COVID-19 patients in wave II were older, had a higher rate of abnormal CXR, dyslipidemia, CCF, and FIO2. Increasing age, underlying cardiovascular diseases, and lower PaO2/FiO2 ratio values comprised the Brixia score; when this score was higher, it was a significant predictor of COVID-19 death [35]. Brixia score on initial CXR predicts fatal outcome in COVID-19 patients (based on in-hospital and out-of-hospital deaths) [35]. The higher proportion of CXR abnormality in wave II may be due to more disease severity than in wave I. This is supported by Fig 2, which compares mortality between waves I and II. This also is consistent with studies that found a new strain of the virus caused more severe infections, as evidenced by abnormal CXR [36–39]. This cohort included patients with moderate to severe COVID-19, with 76% being oxygen dependent and 78% receiving corticosteroid therapy. The cohort had a PaO2/FiO2 ratio of 276mmHg, which is consistent with mild acute respiratory distress syndrome, and 22% of the cohort had acute kidney injury.

A meta-analysis revealed that AKI was a common and serious complication of COVID-19 [40]. AKI was associated with older age and having severe COVID-19 [40]. In patients with COVID-19 complicated by AKI, the risk of dying in the hospital was significantly increased [40, 41]. According to one study, nearly half of COVID-19 patients with ARDS had AKI during their hospital stay [41]. This could explain the high rate of AKI and ARDS complications during wave II, indicating that the Beta variant was associated with increased severity and cytokine storm [12]. Currently, the mechanism of AKI in COVID-19 patients is thought to involve SARS-CoV-2 directly attacking intrinsic renal cells [40]. High ACE2 expression in proximal tubular epithelial cells could be a target for kidney damage. As previously described, high-load SARS-CoV-2 infection causes a cytokine storm in which various inflammatory mediators are released, resulting in ischaemia, hypoxia, fibrosis, and kidney damage [42–47]. Furthermore, COVID-19 that is accompanied by high temperature, shock, dehydration, and hypoxemia, and is managed with antibiotics and other potentially nephrotoxic drugs which may cause AKI [44]. Increased age, diabetes mellitus, and hypertension were also found to cause or worsen the occurrence and progression of AKI in our study. This is consistent with other studies that show an association between AKI and the parameters [28, 48]. AKI and increasing age are also plausible explanations for the uricemia observed in wave II. In addition, Moledina et al [47] found that AKI in COVID-19 may be caused by a combination of some typical AKI risk factors, such as hypotension and volume depletion. AKI could also cause hypotension, volume depletion, and shock, as seen in wave I, where shock was predominant. However, viral septic shock is frequently overlooked in clinical diagnosis; studies have shown that COVID-19 patients developed sepsis, leading to subsequent multiple organ dysfunction [34, 44, 49]. We hypothesize that the high rate of shock described in wave I could be hypovolemic and or sepsis related.

Despite emerging evidence that treatment with empirical administration of antibiotics such as azithromycin and ceftriaxone does not reduce the risk of death in hospitalized COVID-19 patients [50], our study found that antibiotics were used more frequently in wave II than in wave I in the district hospital. This could be attributed to the severity of illness associated with wave II, the level of clinical experience of clinicians in the district hospital (often juniors) combined with the delay in turnaround time in obtaining a positive COVID19 result in patients with a syndrome of a pneumonia requiring oxygen. Clinicians in this setting felt more comfortable in commencing empiric antibiotics. A meta-analysis of 18 RCTs enrolling 2826 patients found that higher use of corticosteroids may reduce mortality in patients with ARDS

[51]. Patients who received a longer course of corticosteroids lived longer than those who received a shorter course [51]. The landmark RECOVERY trial found that there was a 28-day mortality reduction in patients hospitalized with COVID-19 who received corticosteroid therapy [52]. However, in patients who are immunocompromised such as HIV, diabetes mellitus, CKD, and Obesity which is associated with low grade inflammation, high doses and a prolonged course of corticosteroids render COVID-19 patients susceptible to active and latent TB [53]. This could also lead to hypo inflammation and delayed viral clearance, resulting in increased SARS-CoV-2 viral shedding in the lungs and kidneys, explaining ARDS and AKI in wave II. The use of high-flow oxygen through a nasal cannula significantly reduced the need for mechanical ventilation support and the time to clinical recovery in patients with severe COVID-19 when compared to conventional low-flow oxygen therapy [54]. It is worth noting that patients admitted with severe COVID-pneumonia were not given mechanical ventilation or high flow nasal oxygen at the district hospital setting due to limited human and infrastructure resources.

To the best of our knowledge, this is the first relatively large retrospective cohort that provides precise information about the characteristics of noncommunicable disease in waves I and II in terms of demographics, clinical characteristics, triage data and laboratory results, interventions, and complications patterns, including a latent analysis in a high HIV/TB district hospital. This study used data from routine clinical practice; thus, data could be incomplete or incorrect, leading to potential misclassification. To mitigate the impact of incompleteness, we have selectively reported database-specific outcomes. Patients who required mechanical ventilation or high flow nasal oxygen were transferred to our referral centre due to limited resources.

## Conclusion

In conclusion, this study compares demographic, clinical, laboratory, complications, and phenotype differences between COVID-19 waves I and II individuals with NCDs admitted to a district hospital with a high HIV/TB burden. Hypertension, diabetes, HIV/AIDS, obesity, CKD, CCF, COPD, and stroke were the most common comorbidity patterns. Except for HIV status, obesity, and previous stroke, all of them were similar between the two waves. In terms of clinical and laboratory findings, our study found that wave I had statistically significant higher haemoglobin and HIV viral load. However, wave II had statistically significant higher CXR abnormalities, FIO2, and uraemia. The average survival time was longer in wave I, and the average length of stay of patients discharged alive was shorter. The adjusted odds ratio for death vs discharge between waves I and II was not statistically different. LCA revealed that the cardiovascular phenotype was the most likely to die, followed by the diabetes and CKD phenotypes. Diabetes and hypertension phenotypes had the lowest mortality rate. This study, conducted in a district level hospital in South Africa, demonstrates the impact of COVID-19 in low-resource settings. NCDs continue to play a significant role in the outcomes of COVID-19 patients. Additional interventions are required in the treatment of COVID-19 associated with NCDs because this association may cause elevated risk of complications and death among patients attending care at the district hospital level.

## Supporting information

**S1 Dataset.**
(XLSX)

## Acknowledgments

We want to thank the COVID-19 Research Response Collaboration at the Faculty of Medicine and Health Sciences, Stellenbosch University.

## Author Contributions

**Conceptualization:** Ayanda Trevor Mnguni, Denzil Schietekat, Peter S. Nyasulu.

**Data curation:** Ayanda Trevor Mnguni, Denzil Schietekat, Nabilah Ebrahim, Nawhaal Sonday, Nicholas Boliter, Neshaad Schrueder, Shiraaz Gabriels, Annibale Cois, Yamanya Tembo.

**Formal analysis:** Annibale Cois, Yamanya Tembo, Rene English.

**Funding acquisition:** Peter S. Nyasulu.

**Investigation:** Ayanda Trevor Mnguni, Nabilah Ebrahim, Nawhaal Sonday, Nicholas Boliter, Neshaad Schrueder, Shiraaz Gabriels, Jacques L. Tamuzi, Mary-Ann Davies, Peter S. Nyasulu.

**Methodology:** Ayanda Trevor Mnguni, Denzil Schietekat, Nawhaal Sonday, Nicholas Boliter, Neshaad Schrueder, Shiraaz Gabriels, Annibale Cois, Jacques L. Tamuzi, Mary-Ann Davies, Rene English, Peter S. Nyasulu.

**Project administration:** Ayanda Trevor Mnguni, Peter S. Nyasulu.

**Resources:** Peter S. Nyasulu.

**Software:** Annibale Cois, Yamanya Tembo.

**Supervision:** Ayanda Trevor Mnguni, Peter S. Nyasulu.

**Validation:** Ayanda Trevor Mnguni, Nabilah Ebrahim, Nawhaal Sonday, Nicholas Boliter, Neshaad Schrueder, Shiraaz Gabriels, Annibale Cois, Jacques L. Tamuzi, Yamanya Tembo, Mary-Ann Davies, Rene English, Peter S. Nyasulu.

**Visualization:** Nabilah Ebrahim, Annibale Cois, Jacques L. Tamuzi, Yamanya Tembo, Mary-Ann Davies, Rene English, Peter S. Nyasulu.

**Writing – original draft:** Ayanda Trevor Mnguni, Denzil Schietekat, Nabilah Ebrahim, Nawhaal Sonday, Nicholas Boliter, Neshaad Schrueder, Shiraaz Gabriels, Annibale Cois, Jacques L. Tamuzi, Yamanya Tembo, Mary-Ann Davies, Rene English, Peter S. Nyasulu.

**Writing – review & editing:** Ayanda Trevor Mnguni, Denzil Schietekat, Nabilah Ebrahim, Nawhaal Sonday, Nicholas Boliter, Neshaad Schrueder, Shiraaz Gabriels, Annibale Cois, Jacques L. Tamuzi, Yamanya Tembo, Mary-Ann Davies, Rene English, Peter S. Nyasulu.

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
