## [Decision Letter · Decision Letter 0]

2 Dec 2022

PONE-D-22-30534The interface between SARS-CoV-2 and non-communicable diseases (NCDs) in a high HIV/TB burden district level hospital setting, Cape Town, South AfricaPLOS ONE

Dear Dr. Nyasulu,

Thank you for submitting your manuscript to PLOS ONE. After careful consideration, we feel that it has merit but does not fully meet PLOS ONE’s publication criteria as it currently stands. Therefore, we invite you to submit a revised version of the manuscript that addresses the points raised during the review process.

We look forward to receiving your revised manuscript.

Kind regards,

Benjamin M. Liu, MBBS, PhD, D(ABMM), MB(ASCP)

Academic Editor

PLOS ONE

 “The funders had no role in study design, data collection and analysis, decision to publish, or preparation of the manuscript”

“I have read the journal's policy and the authors of this manuscript have the following competing interests”

7. Please include a separate caption for each figure in your manuscript.

8. We note that Figure 1 in your submission contain [map/satellite] images which may be copyrighted. All PLOS content is published under the Creative Commons Attribution License (CC BY 4.0), which means that the manuscript, images, and Supporting Information files will be freely available online, and any third party is permitted to access, download, copy, distribute, and use these materials in any way, even commercially, with proper attribution. For these reasons, we cannot publish previously copyrighted maps or satellite images created using proprietary data, such as Google software (Google Maps, Street View, and Earth). For more information, see our copyright guidelines: http://journals.plos.org/plosone/s/licenses-and-copyright.

Reviewers' comments:

Reviewer's Responses to Questions

**Comments to the Author**

1. Is the manuscript technically sound, and do the data support the conclusions?

Reviewer #1: Yes

2. Has the statistical analysis been performed appropriately and rigorously? 

Reviewer #1: Yes

3. Have the authors made all data underlying the findings in their manuscript fully available?

Reviewer #1: Yes

4. Is the manuscript presented in an intelligible fashion and written in standard English?

Reviewer #1: Yes

5. Review Comments to the Author

Reviewer #1: PLOS one review

Abstract:

Introduction

Line 52: For the sake of consistency, I suggest you use 13.7% instead of 13,7%.

Methods

Line 108: …..admission, from digital registries, with additional data were captured from electronic…

Comment: …with additional data captured from. Delete ‘were’.

Line 113: 30 kg/m2

Comment: It is square, and not 2 (kg/m2)

Line 126: ….and standard deviation (SE)

Comment: Do you mean standard deviation (SD)?

Results

Line 150: Table 1

Comment: I think the ‘N’ column is not necessary. Apart from HIV/AIDS, the rest are 560. I suggest you put asterisk (*) on the HIV/AIDS variable, and define/explain the denominator below the table.

Age class (years)

On line 82 (study population), you mentioned individuals aged 18 years and above were recruited. Why do you have an age group/class from 14 years?? If you think it’s from 14 years, kindly correct that in the study population.

References

Line 435:

Comment: All your on-line references must include dates accessed.

Additional comments:

Is Figure 1 an original map generated by the authors? If no, kindly reference the source.

The title for figure 1 needs to be enhanced. ‘Khayelitsha District in Cape Town/South Africa’.

- I suggest ‘Map of xxx showing xxx District. And please do not combine Cape Town/South Africa like that. I guess they are not the same, so please be specific. If the map is a South African map, say so; if it is a map of Cape Town, let the readers know.

- Again, kindly delete ‘Map of Cape Town/ Khayelitsha District’ on the top left corner of the map.

6. PLOS authors have the option to publish the peer review history of their article (what does this mean?). If published, this will include your full peer review and any attached files.

Reviewer #1: No

---

## [Author Response · Author response to Decision Letter 0]

17 Dec 2022

Reviewers’ responses

Manuscript: The interface between SARS-CoV-2 and non-communicable diseases (NCDs) in a high HIV/TB burden district level hospital setting, Cape Town, South Africa

Editor 

Comment 1: Please ensure that your manuscript meets PLOS ONE's style requirements, including those for file naming. The PLOS ONE style templates can be found at

Response 1: Thank you for this comment. This manuscript has been written in accordance with the PLOS One guideline.

Comment 2: Please provide additional details regarding participant consent. In the ethics statement in the Methods and online submission information, please ensure that you have specified (1) whether consent was informed and (2) what type you obtained (for instance, written or verbal, and if verbal, how it was documented and witnessed). If your study included minors, state whether you obtained consent from parents or guardians. If the need for consent was waived by the ethics committee, please include this information.

Response 2: Thanks. In the tracking manuscript, line 142-145, we have stated the following: “Since the data were obtained from hospital records, a waiver of consent was requested and granted. The data were entered and analyzed anonymously, and no attempt was made to link the data to an individual identifier”.

Comment 3: Thank you for stating the following financial disclosure:

 “The funders had no role in study design, data collection and analysis, decision to publish, or preparation of the manuscript”

Response 3: In the tracking manuscript 452-452, we have stated that “The authors received no specific funding for this work.”

Comment 4: Thank you for stating the following in your Competing Interests section: 

“I have read the journal's policy and the authors of this manuscript have the following competing interests”

Response 4: Thanks. In the tracking manuscript line 449-450, we have stated that "The authors have declared that no competing interests exist."

Comment 5: We note that you have stated that you will provide repository information for your data at acceptance. Should your manuscript be accepted for publication, we will hold it until you provide the relevant accession numbers or DOIs necessary to access your data. If you wish to make changes to your Data Availability statement, please describe these changes in your cover letter and we will update your Data Availability statement to reflect the information you provide.

Response 5: Thanks. In the tracking manuscript line 453-455, the data availability has been changed as follows: “The dataset can be obtained by submitting a request form to the corresponding author.

---

## [Decision Letter · Decision Letter 1]

27 Dec 2022

The interface between SARS-CoV-2 and non-communicable diseases (NCDs) in a high HIV/TB burden district level hospital setting, Cape Town, South Africa

PONE-D-22-30534R1

Dear Dr. Nyasulu,

We’re pleased to inform you that your manuscript has been judged scientifically suitable for publication and will be formally accepted for publication once it meets all outstanding technical requirements.

Kind regards,

Benjamin M. Liu, MBBS, PhD, D(ABMM), MB(ASCP)

Academic Editor

PLOS ONE

Additional Editor Comments (optional):

Reviewers' comments:

Reviewer's Responses to Questions

**Comments to the Author**

1. If the authors have adequately addressed your comments raised in a previous round of review and you feel that this manuscript is now acceptable for publication, you may indicate that here to bypass the “Comments to the Author” section, enter your conflict of interest statement in the “Confidential to Editor” section, and submit your "Accept" recommendation.

Reviewer #1: All comments have been addressed

2. Is the manuscript technically sound, and do the data support the conclusions?

Reviewer #1: Yes

3. Has the statistical analysis been performed appropriately and rigorously? 

Reviewer #1: Yes

4. Have the authors made all data underlying the findings in their manuscript fully available?

Reviewer #1: Yes

5. Is the manuscript presented in an intelligible fashion and written in standard English?

Reviewer #1: Yes

6. Review Comments to the Author

Reviewer #1: The revised version of the manuscript looks good.

The authors have addressed all comments, and suggestions made in the previous manuscript. Thank you.

7. PLOS authors have the option to publish the peer review history of their article (what does this mean?). If published, this will include your full peer review and any attached files.

Reviewer #1: No

---

## [Editor Report · Acceptance letter]

27 Feb 2023

PONE-D-22-30534R1 

The interface between SARS-CoV-2 and non-communicable diseases (NCDs) in a high HIV/TB burden district level hospital setting, Cape Town, South Africa 

Dear Dr. Nyasulu:

I'm pleased to inform you that your manuscript has been deemed suitable for publication in PLOS ONE. Congratulations! Your manuscript is now with our production department. 

Kind regards, 

on behalf of

Dr. Benjamin M. Liu 

Academic Editor

PLOS ONE